# Dependent nonparametric trees for dynamic hierarchical clustering

**Avinava Dubey**[*†], **Qirong Ho**[*‡], **Sinead Williamson**[£], **Eric P. Xing**[†]
† Machine Learning Department, Carnegie Mellon University
‡ Institute for Infocomm Research, A*STAR
£ McCombs School of Business, University of Texas at Austin
akdubey@cs.cmu.edu, hoqirong@gmail.com
sinead.williamson@mccombs.utexas.edu, epxing@cs.cmu.edu

## Abstract

Hierarchical clustering methods offer an intuitive and powerful way to model a wide variety of data sets. However, the assumption of a fixed hierarchy is often overly restrictive when working with data generated over a period of time: We expect both the structure of our hierarchy, and the parameters of the clusters, to evolve with time. In this paper, we present a distribution over collections of time-dependent, infinite-dimensional trees that can be used to model evolving hierarchies, and present an efficient and scalable algorithm for performing approximate inference in such a model. We demonstrate the efficacy of our model and inference algorithm on both synthetic data and real-world document corpora.

## 1 Introduction

Hierarchically structured clustering models offer a natural representation for many forms of data. For example, we may wish to hierarchically cluster animals, where "dog" and "cat" are subcategories of "mammal", and "poodle" and "dachshund" are subcategories of "dog". When modeling scientific articles, articles about machine learning and programming languages may be subcategories under computer science. Representing clusters in a tree structure allows us to explicitly capture these relationships, and allow clusters that are closer in tree-distance to have more similar parameters.

Since hierarchical structures occur commonly, there exists a rich literature on statistical models for trees. We are interested in *nonparametric* distributions over trees – that is, distributions over trees with infinitely many leaves and infinitely many internal nodes. We can model any finite data set using a finite subset of such a tree, marginalizing over the infinitely many unoccupied branches. The advantage of such an approach is that we do not have to specify the tree dimensionality in advance, and can grow our representation in a consistent manner if we observe more data.

In many settings, our data points are associated with a point in time – for example the date when a photograph was taken or an article was written. A stationary clustering model is inappropriate in such a context: The number of clusters may change over time; the relative popularities of clusters may vary; and the location of each cluster in parameter space may change. As an example, consider a topic model for scientific articles over the twentieth century. The field of computer science – and therefore topics related to it – did not exist in the first half of the century. The proportion of scientific articles devoted to genetics has likely increased over the century, and the terminology used in such articles has changed with the development of new sequencing technology.

Despite this, to the best of our knowledge, there are no nonparametric distributions over time-evolving trees in the literature. There exist a variety of distributions over *stationary* trees [1, 14, 5, 13, 10], and time-evolving non-hierarchical clustering models [16, 7, 11, 2, 4, 12] – but no models that combine time evolution and hierarchical structure. The reason for this is likely to be practical: Inference in trees is typically very computationally intensive, and adding temporal variation will, in general, increase the computational requirements. Designing such a model must, therefore, proceed hand in hand with developing efficient and scalable inference schemes.

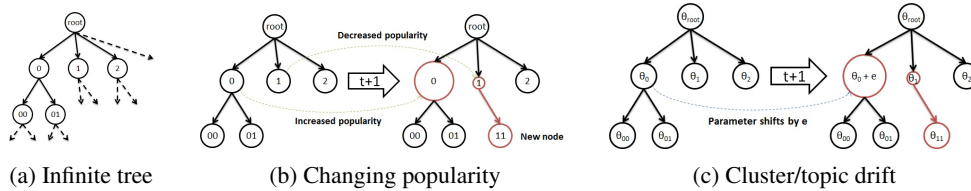

| (a) Infinite tree | (b) Changing popularity | (c) Cluster/topic drift |

Figure 1: Our dependent tree-structured stick breaking process can model trees of arbitrary size and shape, and captures popularity and parameter changes through time. **a)** Model any number of nodes (clusters, topics), of any branching factor, and up to any depth **b)** Nodes can change in probability mass, or new nodes can be created **c)** Node parameters can evolve over time.

In this paper, we define a distribution over temporally varying trees with infinitely many nodes that captures this form of variation, and describe how this model can cluster both real-valued observations and text data. Further, we propose a scalable approximate inference scheme that can be run in parallel, and demonstrate its efficacy on synthetic data where ground-truth clustering is available, as well as demonstrate qualitative and quantitative performance on three text corpora.

## 2 Background

The model proposed in this paper is a dependent nonparametric process with tree-structured marginals. A dependent nonparametric process [12] is a distribution over collections of random measures indexed by values in some covariate space, such that at each covariate value, the marginal distribution is given by some known nonparametric distribution. For example, a dependent Dirichlet process [12, 7, 11] is a distribution over collections of probability measures with Dirichlet process-distributed marginals; a dependent Pitman-Yor process [15] is a distribution over collections of probability measures with Pitman-Yor process-distributed marginals; a dependent Indian buffet process [17] is a distribution over collections of matrices with Indian buffet process-distributed marginals; etc. If our covariate space is time, such distributions can be used to construct nonparametric, time-varying models.

There are two main methods of inducing dependency: Allowing the sizes of the atoms composing the measure to vary across covariate space, and allowing the parameter values associated with the atoms to vary across covariate space. In the context of a time-dependent topic model, these methods correspond to allowing the popularity of a topic to change over time, and allowing the words used to express a topic to change over time (topic drift). Our proposed model incorporates both forms of dependency. In the supplement, we discuss some specific dependent nonparametric models that share properties with our model.

The key difference between our proposed model and existing dependent nonparametric models is that ours has tree-distributed marginals. There are a number of options for the marginal distribution over trees, as we discuss in the supplement. We choose a distribution over infinite-dimensional trees known as the tree-structured stick breaking process [TSSBP, 1], described in Section 2.1.

### 2.1 The tree-structured stick-breaking process

The tree-structured stick-breaking process (TSSBP) is a distribution over trees with infinitely many leaves and infinitely many internal nodes. Each node $\epsilon$ within the tree is associated with a mass $\pi_\epsilon$ such that $\sum_\epsilon \pi_\epsilon = 1$, and each data point is assigned to a node in the tree according to $p(z_n = \epsilon) = \pi_\epsilon$, where $z_n$ is the node assignment of the $n$th data point. The TSSBP is unique among the current toolbox of random infinite-dimensional trees in that data can be assigned to an internal node, rather than a leaf, of the tree. This property is often desirable; for example in a topic modeling context, a document could be assigned to a general topic such as "science" that lives toward the root of the tree, or to a more specific topic such as "genetics" that is a descendant of the science topic.

The TSSBP can be represented using two interleaving stick-breaking processes – one (parametrized by $\alpha$) that determines the size of a node and another (parametrized by $\gamma$) that determines the branching probabilities. Index the root node as node $\emptyset$ and let $\pi_\emptyset$ be the mass assigned to it. Index its (countably infinite) child nodes as node 1, node 2, ... and let $\pi_1, \pi_2, \ldots$ be the masses assigned to them; index the child nodes of node 1 as nodes $1 \cdot 1, 1 \cdot 2, \ldots$ and let $\pi_{1 \cdot 1}, \pi_{1 \cdot 2}, \ldots$ be the masses assigned to nodes $1 \cdot 1, 1 \cdot 2 \ldots$; etc. Then we can sample the infinite-dimensional tree as:

$$\nu_\epsilon \sim \text{Beta}(1, \alpha(|\epsilon|)), \quad \psi_\epsilon \sim \text{Beta}(1, \gamma), \quad \pi_\emptyset = \nu_\emptyset, \quad \phi_\emptyset = 1$$
$$\phi_{\epsilon \cdot i} = \psi_{\epsilon \cdot i} \prod_{j=1}^{i-1} (1 - \psi_{\epsilon \cdot j}) \quad \pi_\epsilon = \nu_\epsilon \phi_\epsilon \prod_{\epsilon' \prec \epsilon} (1 - \nu_{\epsilon'}) \phi_{\epsilon'}, \tag{1}$$

where $|\epsilon|$ indicates the depth of node $\epsilon$, and $\epsilon' \prec \epsilon$ indicates that $\epsilon'$ is an ancestor node of $\epsilon$. We refer to the resulting infinite-dimensional weighted tree as $\Pi = ((\pi_\epsilon), (\phi_{\epsilon i}))$.

# 3 Dependent tree-structured stick-breaking processes

We now describe a dependent tree-structured stick-breaking process where both atom sizes and their locations vary with time. We first describe a distribution over atom sizes, and then use this distribution over collections of trees as the basis for time-varying clustering models and topic models.

## 3.1 A distribution over time-varying trees

We start with the basic TSSBP model [1] (described in Section 2.1 and the left of Figure 1), and modify it so that the latent variables $\nu_\epsilon$, $\psi_\epsilon$ and $\pi_\epsilon$ are replaced with sequences $\nu_\epsilon^{(t)}$, $\psi_\epsilon^{(t)}$ and $\pi_\epsilon^{(t)}$ indexed by discrete time $t \in \mathcal{T}$ (the middle of Figure 1). The forms of $\nu_\epsilon^{(t)}$ and $\psi_\epsilon^{(t)}$ are chosen so that the marginal distribution over the $\pi_\epsilon^{(t)}$ is as described in Equation 1.

Let $N^{(t)}$ be the number of observations at time $t$, and let $z_n^{(t)}$ be the node allocation of the $n$th observation at time $t$. For each node $\epsilon$ at time $t$, let $X_\epsilon^{(t)} = \sum_{n=1}^{N_t} \mathbb{I}(z_n^{(t)} = \epsilon)$ be the number of observations assigned to node $\epsilon$ at time $t$, and $Y_\epsilon^{(t)} = \sum_{n=1}^{N_t} \mathbb{I}(\epsilon \prec z_n^{(t)})$ be the number of observations assigned to descendants of node $\epsilon$. Introduce a "window" parameter $h \in \mathbb{N}$. We can then define a prior predictive distribution over the tree at time $t$, as

$$
\begin{aligned}
\nu_\epsilon^{(t)} &\sim \text{Beta}\big(1 + \textstyle\sum_{t'=t-h}^{t-1} X_\epsilon^{(t')}, \alpha(|\epsilon|) + \sum_{t'=t-h}^{t-1} Y_\epsilon^{(t')}\big) \\
\psi_{\epsilon \cdot i}^{(t)} &\sim \text{Beta}\big(1 + \textstyle\sum_{t'=t-h}^{t-1}(X_{\epsilon \cdot i}^{(t')} + Y_{\epsilon \cdot i}^{(t')}), \gamma + \sum_{j>i} \sum_{t'=t-h}^{t}(X_{\epsilon \cdot j}^{(t')} + Y_{\epsilon \cdot j}^{(t')})\big).
\end{aligned}
\tag{2}
$$

Following [1], we let $\alpha(d) = \lambda^d \alpha_0$, for $\alpha_0 > 0$ and $\lambda \in (0, 1)$. This defines a sequence of trees $(\Pi^{(t)} = ((\pi_\epsilon^{(t)}), (\phi_{\epsilon i}^{(t)})), t \in \mathcal{T})$.

Intuitively, the prior distribution over a tree at time $t$ is given by the posterior distribution of the (stationary) TSSBP, conditioned on the observations in some window $t - h, \ldots, t - 1$. The following theorem gives the equivalence of dynamic TSSBP (dTSSBP) and TSSBP

**Theorem 1.** *The marginal posterior distribution of the dTSSBP, at time $t$, follows a TSSBP.*

The proof is a straightforward extension of that for the generalized Pólya urn dependent Dirichlet process [7] and is given in the supplimentary. The above theorem implies that Equation 2 defines a *dependent tree-structured stick-breaking process*.

We note that an alternative choice for inducing dependency would be to down-weight the contribution of observations for previous time-steps. For example, we could exponentially decay the contributions of observations from previous time-steps, inducing a similar form of dependency as that found in the recurrent Chinese restaurant process [2]. However, unlike the method described in Equation 2, such an approach would not yield stationary TSSBP-distributed marginals.

## 3.2 Dependent hierarchical clustering

The construction above gives a distribution over infinite-dimensional trees, which in turn have a probability distribution over their nodes. In order to use this distribution in a hierarchical Bayesian model for data, we must associate each node with a parameter value $\theta_\epsilon^{(t)}$. We let $\Theta^{(t)}$ denote the set of all parameters $\theta_\epsilon^{(t)}$ associated with a tree $\Pi^{(t)}$. We wish to capture two properties: 1) Within a tree $\Pi^{(t)}$, nodes have similar values to their parents; and 2) Between trees $\Pi^{(t)}$ and $\Pi^{(t+1)}$, corresponding parameters $\theta_\epsilon^{(t)}$ and $\theta_\epsilon^{(t+1)}$ have similar values. This form of variation is shown in the right of Figure 1. In this subsection, we present two models that exhibit these properties: One appropriate for real-valued data, and one appropriate for multinomial data.

### 3.2.1 A time-varying, tree-structured mixture of Gaussians

An infinite mixture of Gaussians is a flexible choice for density estimation and clustering real-valued observations. Here, we suggest a time-varying hierarchical clustering model that is similar to the generalized Gaussian model of [1]. The model assumes Gaussian-distributed data at each node, and allows the means of clusters to evolve in an auto-regressive model, as below:

$$
\theta_\emptyset^{(t)} | \theta_\emptyset^{(t-1)} \sim \mathcal{N}(\theta_\emptyset^{(t-1)}, \sigma_0 \sigma_1^a \mathbf{I}), \quad \theta_{\epsilon \cdot i}^{(t)} | \theta_\epsilon^{(t)}, \theta_{\epsilon \cdot i}^{(t-1)} \sim \mathcal{N}(m, s^2 \mathbf{I}),
\tag{3}
$$

where, $s^2 = \left( \frac{1}{\sigma_0 \sigma_1^{|\epsilon \cdot i|}} + \frac{1}{\sigma_0 \sigma_1^{|\epsilon \cdot i| + a}} \right)^{-1}$, $\quad m = s^2 \cdot \left( \frac{\theta_\epsilon^{(t)}}{(\sigma_0 \sigma_1^{|\epsilon \cdot i|})^2} + \frac{\eta \theta_{\epsilon \cdot i}^{(t-1)}}{\sigma_0 \sigma_1^{|\epsilon \cdot i| + a}} \right)$, $\sigma_0 > 0$, $\sigma_1 \in (0, 1)$, $\eta \in [0, 1)$, and $a \geq 1$. Due to the self-conjugacy of the Gaussian distribution, this corresponds to a Markov network with factor potentials given by unnormalized Gaussian distributions: Up to a normalizing constant, the factor potential associated with the link between $\theta_\epsilon^{(t-1)}$ and $\theta_\epsilon^{(t)}$ is Gaussian with variance $\sigma_0 \sigma_1^{|\epsilon|}$, and the factor potential associated with the link between $\theta_\epsilon^{(t)}$ and $\theta_{\epsilon \cdot i}^{(t)}$ is Gaussian with variance $\sigma_0 \sigma_1^{|\epsilon \cdot i| + a}$.

For a single time point, this allows for fractal-like behavior, where the distance between child and parent decreases down the tree. This behavior, which is not used in the generalized Gaussian model of [1], makes it easier to identify the root node, and guarantees that the marginal distribution over the location of the leaf nodes has finite variance. The $a$ parameter enforces the idea that the amount of variation between $\theta_\epsilon^{(t)}$ and $\theta_\epsilon^{(t+1)}$ is smaller than that between $\theta_\epsilon^{(t)}$ and $\theta_{\epsilon \cdot i}^{(t)}$, while $\eta$ ensures the variance of node parameters remains finite across time. We chose spherical Gaussian distributions to ensure that structural variation is captured by the tree rather than by node parameters.

### 3.3 A time-varying model for hierarchically clustering documents

Given a dictionary of $V$ words, a document can be represented using a $V$-dimensional term frequency vector, that corresponds to a location on the surface of the $(V-1)$-dimensional unit sphere. The von Mises-Fisher distribution, with mean direction $\boldsymbol{\mu}$ and concentration parameter $\tau$, provides a distribution on this space. A mixture of von Mises-Fisher distributions can, therefore, be used to cluster documents [3, 8]. Following the terminology of topic modeling [6], the mean direction $\boldsymbol{\mu}_k$ associated with the $k$th cluster can be interpreted as the topic associated with that cluster.

We construct a time-dependent hierarchical clustering model appropriate for documents by associating nodes of our dependent nonparametric tree with topics. Let $\mathbf{x}_n^{(t)}$ be the vector associated with the $n$th document at time $t$. We assign a mean parameter $\theta_\epsilon^{(t)}$ to each node $\epsilon$ in each tree $\Pi^{(t)}$ as

$$\theta_\emptyset^{(t)} | \theta_\emptyset^{(t-1)} \sim \text{vMF}(\tau_\emptyset^{(t)}, \rho_\emptyset^{(t)}), \quad \theta_{\epsilon \cdot i}^{(t)} | \theta_\epsilon^{(t)}, \theta_{\epsilon \cdot i}^{(t-1)} \sim \text{vMF}(\tau_{\epsilon \cdot i}^{(t)}, \rho_{\epsilon \cdot i}^{(t)}), \tag{4}$$

where, $\rho_\emptyset^{(t)} = \kappa_0 \sqrt{1 + \kappa_1^{2a} + 2\kappa_1^a (\theta_{-1}^{(t)} \cdot \theta_\emptyset^{(t-1)})}$, $\quad \tau_\emptyset^{(t)} = \frac{\kappa_0 \theta_{-1}^{(t)} + \kappa_0 \kappa_1^a \theta_\emptyset^{(t-1)}}{\rho_\emptyset^{(t)}}$ $\quad \rho_{\epsilon \cdot i}^{(t)} = \kappa_0 \kappa_1^{|\epsilon \cdot i|} \sqrt{1 + \kappa_1^{2a} + 2\kappa_1^a (\theta_\epsilon^{(t)} \cdot \theta_{\epsilon \cdot i}^{(t-1)})}$, $\quad \tau_{\epsilon \cdot i}^{(t)} = \frac{\kappa_0 \kappa_1^{|\epsilon \cdot i|} \theta_\epsilon^{(t)} + \kappa_0 \kappa_1^{|\epsilon \cdot i| + a} \theta_{\epsilon \cdot i}^{(t-1)}}{\rho_{\epsilon \cdot i}^{(t)}}$, $\kappa_0 > 0$, $\kappa_1 > 1$, and $\theta_{-1}^{(t)}$ is a probability vector of the same dimension as the $\theta_\epsilon^{(t)}$ that can be interpreted as the parent of the root node at time $t$.[1] This yields similar dependency behavior to that described in Section 3.2.1.

Conditioned on $\Pi^{(t)}$ and $\Theta^{(t)} = (\theta_\epsilon^{(t)})$, we sample each document $\mathbf{x}_n^{(t)}$ according to $z_n^{(t)} \sim$ Discrete($\Pi^{(t)}$) and $\mathbf{x}_n \sim \text{vMF}(\theta^{(t)}, \beta)$. This is a hierarchical extension of the temporal vMF mixture proposed by [8].

## 4 Online Learning

In many time-evolving applications, we observe data points in an online setting. We are typically interested in obtaining predictions for future data points, or characterizing the clustering structure of current data, rather than improving predictive performance on historic data. We therefore propose a sequential online learning algorithm, where at each time $t$ we infer the parameter settings for the tree $\Pi^{(t)}$ conditioned on the previous trees, which we do not re-learn. This allows us to focus our computational efforts on the most recent (and likely relevant) data. This has the added advantage of reducing the computational demands of the algorithm, as we do not incorporate a backwards pass through the data, and are only ever considering a fraction of the data at a time.

In developing an inference scheme, there is always a trade-off between estimate quality and computational requirements. MCMC samplers are often the "gold standard" of inference techniques, because they have the true posterior distribution as the stationary distribution of their Markov Chain. However, they can be very slow, particularly in complex models. Estimating the parameter setting that maximizes the data likelihood is a much cheaper, but cannot capture the full posterior.

In order to develop an inference algorithm that is parallelizable, runs in reasonable time, but still obtains good predictive performance, we combine Gibbs sampling steps for learning the tree parameters ($\Pi^{(t)}$) and the topic indicators ($z_n^{(t)}$) with a MAP method for estimating the location parameters ($\theta_\epsilon^{(t)}$). The resulting algorithm has the following desirable properties:

1. The priors for $\nu_\epsilon^{(t)}, \psi_\epsilon^{(t)}$ only depend on $\{z_n^{(0)}\} \ldots \{z_n^{(t-1)}\}$, whose sufficient statistics $\{X_\epsilon^{(0)}, Y_\epsilon^{(0)}\} \ldots \{X_\epsilon^{(t-1)}, Y_\epsilon^{(t-1)}\}$ can be updated in amortized constant time.

2. The posteriors for $\nu_\epsilon^{(t)}, \psi_\epsilon^{(t)}$ are conditionally independent given $\{z_n^{(1)}\} \ldots \{z_n^{(t)}\}$. Hence we can Gibbs sample $\nu_\epsilon^{(t)}, \psi_\epsilon^{(t)}$ in parallel given the cluster assignments $\{z_n^{(1)}\} \ldots \{z_n^{(t)}\}$ (or more precisely, their sufficient statistics $\{X_\epsilon, Y_\epsilon\}$). Similarly, we can Gibbs sample the cluster/topic assignments $\{z_n^{(t)}\}$ in parallel given the parameters $\{\nu_\epsilon^{(t)}, \psi_\epsilon^{(t)}, \theta_\epsilon^{(t)}\}$ and the data, as well as infer the MAP estimate of $\{\theta_\epsilon^{(t)}\}$ in parallel given the data and the cluster/topic assignments. Because of the online assumption, we do not consider evidence from times $u > t$.

**Sampling $\nu_\epsilon^{(t)}, \psi_\epsilon^{(t)}$**   Due to the conjugacy between the beta and binomial distributions, we can easily Gibbs sample the stick-breaking parameters

$$\nu_\epsilon^{(t)} | X_\epsilon, Y_\epsilon \sim \text{Beta}\big(1 + \sum_{t'=t-h}^{t} X_\epsilon^{(t')}, \alpha(|\epsilon|) + \sum_{t'=t-h}^{t} Y_\epsilon^{(t')}\big)$$

$$\psi_{\epsilon \cdot i}^{(t)} | X_{\epsilon \cdot i}, Y_{\epsilon \cdot i} \sim \text{Beta}\big(1 + \sum_{t'=t-h}^{t}(X_{\epsilon \cdot i}^{(t')} + Y_{\epsilon \cdot i}^{(t')}), \gamma + \sum_{j>i} \sum_{t'=t-h}^{t}(X_{\epsilon \cdot j}^{(t')} + Y_{\epsilon \cdot j}^{(t')})\big).$$

The $\nu_\epsilon^{(t)}, \psi_\epsilon^{(t)}$ distributions for each node are conditionally independent given the counts $X, Y$, and so the sampler can be parallelized. We only explicitly store $\pi_\epsilon^{(t)}, \phi_\epsilon^{(t)}, \theta_\epsilon^{(t)}$ for nodes $\epsilon$ with nonzero counts, i.e. $\sum_{t'=t-h}^{t} X_\epsilon^{(t')} + Y_\epsilon^{(t')} > 0$.

**Sampling $z_n^{(t)}$**   Conditioned on the $\nu_\epsilon^{(t)}$ and $\psi_\epsilon^{(t)}$, the distribution over the cluster assignments $z_n^{(t)}$ is just given by the TSSBP. We therefore use the slice sampling method described in [1] to Gibbs sample $z_n^{(t)} \mid \{\nu_\epsilon^{(t)}\}, \{\psi_\epsilon^{(t)}\}, x_n^{(t)}, \theta$. Since the cluster assignments are conditionally independent given the tree, this step can be performed in parallel.

**Learning $\theta$**   It is possible to Gibbs sample the cluster parameters $\theta$; however, in the document clustering case described in Section 3.3, this requires far more time than sampling all other parameters. To improve the speed of our algorithm, we instead use *maximum a posteriori* (MAP) estimates for $\theta$, obtained using a parallel coordinate ascent algorithm. Notably, conditioned on the trees at time $t-1$ and $t+1$, the $\theta_\epsilon^{(t)}$ for odd-numbered tree depths $|\epsilon|$ are conditionally independent given the $\theta_{\epsilon'}^{(t)}$s at even-numbered tree depths $|\epsilon'|$, and vice versa. Hence, our algorithm alternates between parallel optimization of odd-depth $\theta_\epsilon^{(t)}$, and parallel optimization of even-depth $\theta_\epsilon^{(t)}$.

In general, the conditional distribution of a cluster parameter $\theta_\epsilon^{(t)}$ depends on the values of its predecessor $\theta_\epsilon^{(t-1)}$, its postdecessor $\theta_\epsilon^{(t+1)}$, its parent at time $t$, and its children at time $t$. In some cases, not all of these values will be available – for example if a node was unoccupied at previous time steps. In this case, the distribution now depends on the full history of the parent node. For computational reasons, and because we do not wish to store the full history, we approximate the distribution as being dependent only on observed members of the node's Markov blanket.

## 5   Experimental evaluation

We evaluate the performance of our model on both synthetic and real-world data sets. Evaluation on synthetic data sets allows us to verify that our inference algorithm allows us to recover the "true" evolving hierarchical structure underlying our data. Evaluation on real-world data allows us to evaluate whether our modeling assumptions are useful in practice.

### 5.1   Synthetic data

We manually created a time-evolving tree, as shown in Figure 2, with Gaussian-distributed data at each node. This synthetic time-evolving tree features temporal variation in node probabilities, temporal variation in node parameters, and addition and deletion of nodes. Using the Gaussian model described in Equation 3, we inferred the structure of the tree at each time period as described in Section 4. Figure 3 shows the recovered tree structure, demonstrating the ability of our inference algorithm to recover the expected evolving hierarchical structure. Note that it accurately captures evolution in node probabilities and location, and the addition and deletion of new nodes.

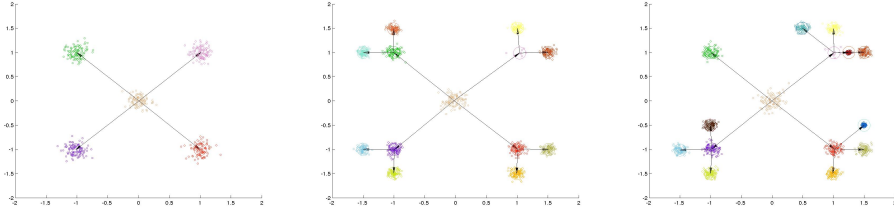

Figure 2: **Ground truth tree**, evolving over three time steps

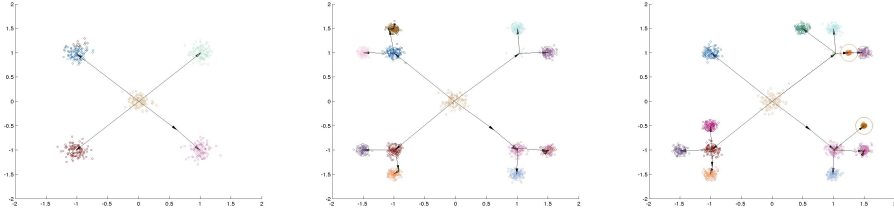

Figure 3: **Recovered tree structure**, over three consecutive time periods. Each color indicates a node in the tree and each arrow indicates a branch connecting parent to child; nodes are consistently colored across time.

| | dTSSBP | | o-TSSBP | | T-TSSBP | |
|---|---|---|---|---|---|---|
| Depth limit | 4 | 3 | 4 | 3 | 4 | 3 |
| TWITTER | **522 ± 4.35** | 249 ± 0.98 | 414 ± 3.31 | 199 ± 2.19 | 335 ± 54.8 | 182 ± 24.1 |
| SOU | **2708 ± 32.0** | 1320 ± 33.6 | 1455 ± 44.5 | 583 ± 16.4 | 1687 ± 329 | 1089 ± 143 |
| PNAS | **4562 ± 116** | 3217 ± 195 | 2672 ± 357 | 1163 ± 196 | 4333 ± 647 | 2962 ± 685 |

| | dDP | o-DP | T-DP |
|---|---|---|---|
| TWITTER | 204 ± 8.82 | 136 ± 0.42 | 112 ± 10.9 |
| SOU | 834 ± 51.2 | 633 ± 18.8 | 890 ± 70.5 |
| PNAS | 2374 ± 51.7 | 1061 ± 10.5 | 2174 ± 134 |

Table 1: Test set average log-likelihood on three datasets.

## 5.2 Real-world data

In Section 3.3, we described how the dependent TSSBP can be combined with a von Mises-Fisher likelihood to cluster documents. To evaluate this model, we looked at three corpora:

- TWITTER: 673,102 tweets containing hashtags relevant to the NFL, collected over 18 weeks in 2011 and containing 2,636 unique words (after stopwording). We grouped the tweets into 9 two-week epochs.
- PNAS: 79,800 paper titles from the Proceedings of the National Academy of Sciences between 1915 and 2005, containing 36,901 unique words (after stopwording). We grouped the titles into 10 ten-year epochs.
- STATE OF THE UNION (SOU): Presidential SoU addresses from 1790 through 2002, containing 56,352 sentences and 21,505 unique words (after stopwording). We grouped the sentences into 21 ten-year epochs.

In each case, documents were represented using their vector of term frequencies.

Our hypothesis is that the topical structure of language is *hierarchically structured* and *time-evolving*, and that a model that captures these properties will achieve better performance than models that ignore hierarchical structure and/or temporal evolution. To test these hypotheses, we compare our dependent tree-structured stick-breaking process (dTSSBP) against several online nonparametric models for document clustering:

1. Multiple tree-structured stick-breaking process (T-TSSBP): We modeled the entire corpus using the stationary TSSBP model, with each node modeled using an independent von Mises-Fisher distribution. Each time period is modeled with a separate tree, using a similar implementation to our time-dependent TSSBP.
2. "Online" tree-structured stick-breaking processes (o-TSSBP): This simulates online learning of a single, stationary tree over the entire corpus. We used our dTSSBP implementation with an infinite window $h = \infty$, and once a node is created at time $t$, we prevent its vMF mean $\theta_\epsilon^{(t)}$ from changing in future time points.
3. Dependent Dirichlet process (dDP): We modeled the entire corpus using an h-order Markov generalized Pólya urn DDP [7]. This model was implemented by modifying our dTSSBP code to have a single level. Node parameters were evolved as $\theta_k^{(t)} \sim \text{vMF}(\theta_k^{(t)}, \xi)$.
4. Multiple Dirichlet process (T-DP): We modeled the entire corpus using DP mixtures of von Mises-Fisher distributions, one DP per time period. Each node was modeled using an independent von Mises-Fisher distribution. We used our own implementation.

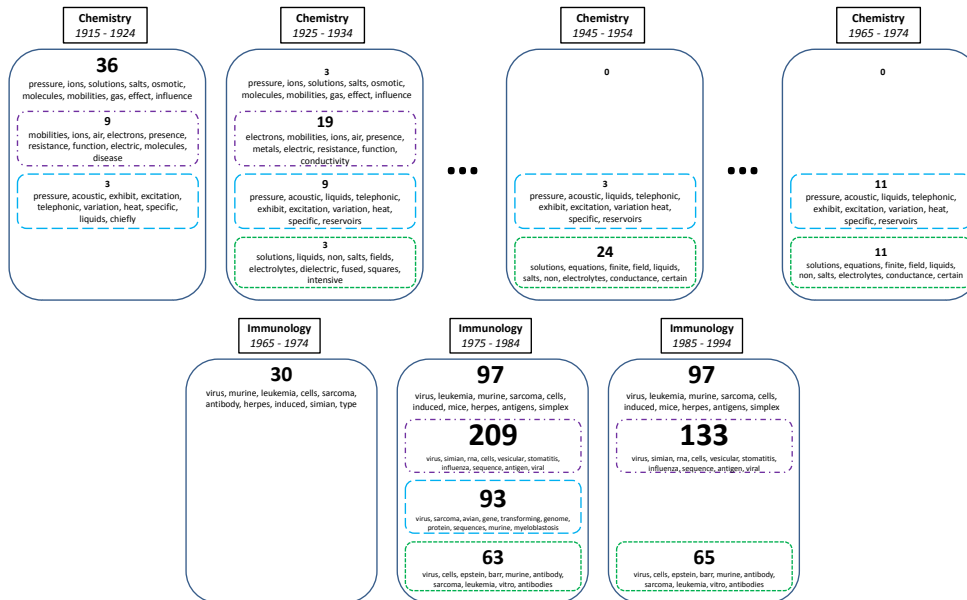

Figure 4: **PNAS dataset:** Birth, growth, and death of tree-structured topics in our dTSSBP model. This illustration captures some trends in American scientific research throughout the 20th century, by focusing on the evolution of parent and child topics in two major scientific areas: Chemistry and Immunology (the rest of the tree has been omitted for clarity). At each epoch, we show the number of documents assigned to each topic, as well as it's most popular words (according to the vMF mean $\theta$).

5. "Online" Dirichlet process (o-DP): This simulates online learning of a single DP over the entire corpus. We used our dDP implementation with an infinite window $h = \infty$, and once a cluster is instantiated at time $t$, we prevent its vMF mean $\theta^{(t)}$ from changing in future time points.

**Evaluation scheme:** We divide each dataset into two parts: the first $50\%$, and last $50\%$ of time points. We use the first $50\%$ to tune model parameters and select a good random restart (by training on $90\%$ and testing on $10\%$ of the data at each time point), and then use the last $50\%$ to evaluate the performance of the best parameters/restart (again, by training on $90\%$ and testing on $10\%$ data). When training the 3 TSSBP-based models, we grid-searched $\kappa_0 \in \{1, 10, 100, 1000, 10000\}$, and fixed $\kappa_1 = 1$, $a = 0$ for simplicity. Each value of $\kappa_0$ was run 5 times to get different random restarts, and we took the best $\kappa_0$-restart pair for evaluation on the last $50\%$ of time points. For the 3 DP-based models, there is no $\kappa_0$ parameter, so we simply took 5 random restarts and used the best one for evaluation. For all TSSBP- and DP-based models, we repeated the evaluation phase 5 times to get error bars. Every dTSSBP trial completed in $< 20$ minutes on a single processor core, while we observed moderate (though not perfectly linear) speedups with 2-4 processors.

**Parameter settings:** For all models, we estimated each node/cluster's vMF concentration parameter $\beta$ from the data. For the TSSBP-based models, we used stick breaking parameters $\gamma = 0.5$ and $\alpha(d) = 0.5^d$, and set $\theta^{(t)}_{-1}$ to the average document term frequency vector at time $t$. In order to keep running times reasonable, we limit the TSSBP-based models to a maximum depth of either 3 or 4 (we report results for both)[2]. For the DP-based models, we used a Dirichlet process concentration parameter of 1. The dDP's inter-epoch vMF concentration parameter was set to $\xi = 0.001$.

**Results:** Table 1 shows the average log (unnormalized) likelihoods on the test sets (from the last $50\%$ of time points). The tree-based models uniformly out-perform the non-hierarchical models, while the max-depth-4 tree models outperform the max-depth-3 ones. On all 3 datasets, the max-depth-4 dTSSBP uniformly outperforms all models, confirming our initial hypothesis.

### 5.3 Qualitative results

In addition to high-quality quantitative results, we find that the time-dependent tree model gives good qualitative performance. Figure 4 shows two time-evolving sub-trees obtained from the PNAS data set. The top level shows a sub-tree concerned with Chemistry; the bottom level shows a sub-tree

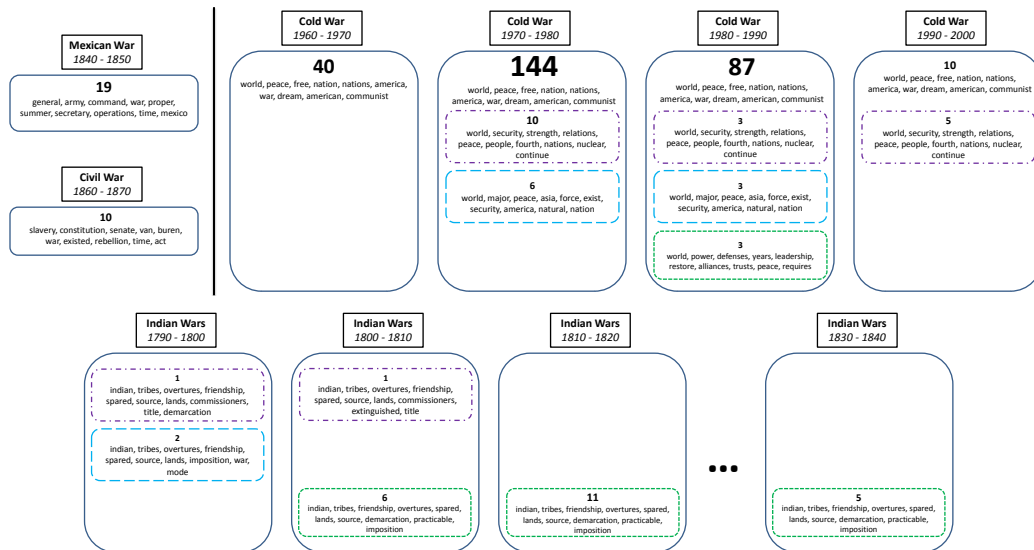

Figure 5: **State of the Union dataset:** Birth, growth, and death of tree-structured topics in our dTSSBP model. This illustration captures some key events in American history. At each epoch, we show the number of documents assigned to each topic, as well as it's most popular words (according to the vMF mean $\theta$).

concerned with Immunology. Our dynamic tree model discovers closely-related topics and groups them under a sub-tree, and creates, grows and destroys individual sub-topics as needed to fit the data. For instance, our model captures the sudden surge in Immunology-related research from 1975-1984, which happened right after the structure of the antibody molecule was identified a few years prior.

In the Chemistry topic, the study of mechanical properties of materials (pressure, acoustic properties, specific heat, etc) is a constant presence throughout the century. The study of electrical properties of materials starts off with a topic (in purple) that seems devoted to Physical Chemistry. However, following the development of Quantum Mechanics in the 30s, this line of research became more closely aligned with Physics than Chemistry, and it disappears from the sub-tree. In its wake, we see the growth of a topic more concerned with electrolytes, solutions and salts, which remained the within the sphere of Chemistry.

Figure 5 shows time-evolving sub-trees obtained from the State of the Union dataset. We see a sub-tree tracking the development of the Cold War. The parent node contains general terms relevant to the Cold War; starting from the 1970s, a child node (shown in purple) contains terms relevant to nuclear arms control, in light of the Strategic Arms Limitation Talks of that decade. The same decade also sees the birth of a child node focused on Asia (shown in cyan), contemporaneous with President Richard Nixon's historic visit to China in 1972. In addition to the Cold War, we also see topics corresponding to events such as the Mexican War, the Civil War and the Indian Wars, demonstrating our model's ability to detect events in a timeline.

# 6 Discussion

In this paper, we have proposed a flexible nonparametric model for dynamically-evolving, hierarchically structured data. This model can be applied to multiple types of data using appropriate choices of likelihood; we present an application in document clustering that combines high-quality quantitative performance with intuitively interpretable results. One of the significant challenges in constructing nonparametric dependent tree models is the need for efficient inference algorithms. We make judicious use of approximations and combine MCMC and MAP approximation techniques to develop an inference algorithm that can be applied in an online setting, while being parallelizable.

**Acknowledgements:** This research was supported by *NSF Big data IIS1447676, DARPA XDATA FA87501220324* and *NIH GWAS R01GM087694*.

## Footnotes

[1] In our experiments, we set $\theta_{-1}^{(t)}$ to be the average over all data points at time $t$. This ensures that the root node is close to the centroid of the data, rather than the periphery.

[2]One justification is that shallow hierarchies are easier to interpret than deep ones; see [5, 9].

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
