[Supplementary Material · supplement.pdf]

# Dependent nonparametric trees for dynamic hierarchical clustering: Supplementary material

**Avinava Dubey**[*†], **Qirong Ho**[*‡], **Sinead Williamson**[£], **Eric P. Xing**[†]

† Machine Learning Department, Carnegie Mellon University
‡ Institute for Infocomm Research, A*STAR
£ McCombs School of Business, University of Texas at Austin
akdubey@cs.cmu.edu, hoqirong@gmail.com
sinead.williamson@mccombs.utexas.edu, epxing@cs.cmu.edu

## 1 Other related work

In this paper, we propose a method for a temporally varying, tree-structured clustering model with an unbounded number of clusters. A number of existing models incorporate one or more of these features.

There exist a wide variety of distributions over trees with infinitely many nodes, including the nested Chinese restaurant process (Blei et al., 2004), the Dirichlet diffusion tree (Neal, 2003), and Kingman's coalescent (Kingman, 1982). These models differ from the TSSBP in that data can only be associated with a leaf node, or equivalently a full path from root to leaf. We chose to base our clustering model on the TSSBP because, in many applications, it makes sense to associate data with internal nodes. For example, a document may be narrowly about Physics or Biology, or may be a more broad article on the sciences in general.

While, to the best of our knowledge, there exist no temporally varying nonparametric tree distributions, there do exist a wide variety of temporally varying nonparametric clustering models, several of which are related to the model proposed in this paper. The dependent Dirichlet process models of Caron et al. (2007) and Lin et al. (2010) specify a distribution over clusterings of data, where the popularity of a cluster can vary over time. These models are based on the Chinese restaurant process: the probability of joining a cluster at time $t$ depends on both the number of words associated with that topic at time $t$ (as in the standard Chinese restaurant process), *and* on the word counts from previous time periods. We modify this approach to allow the node weights in our sequence of trees to vary over time.

Other models have been used to allow the parameters associated with clusters to vary over time. The single-p dependent Dirichlet process (MacEachern, 1999) clusters data according to a Dirichlet process, and evolves the cluster parameters according to a stochastic process. In a parametric setting that is similar to the topic model proposed in Section 3.3, the Dynamic Topic Model (Blei and Lafferty, 2006), parametrizes each topic, or cluster, using a logistic normal distribution. Time dependence is induced by allowing the underlying Gaussian-distributed vector to evolve via multivariate increments.

## 2 Dependent tree-structured stick-breaking process: Marginal properties

One of our key desiderata in constructing a dependent tree is that the marginal distribution at time $t$ is given by an existing nonparametric tree model – in our case, the TSSBP. This allows us to leverage properties of the related stationary model, and avoids explicit dependence on the time at which we first started recording data. The construction described in the main paper exhibits this marginal property:

**Theorem 1.** *The marginal posterior distribution of the dTSSBP, at time t, follows a TSSBP.*

*Proof.* The exchangeable distribution over partitions associated with a Dirichlet process is described using the Ewen's Sampling Formula (ESF). As shown by Caron et al. (2007), the resulting random partition still follows an ESF if, at time $t$ we deterministically delete observations from time $t - h$. The associated posterior random measure at time $t$ will therefore follow a Dirichlet process, following de Finetti's theorem.

This result extends trivially to the dTSSBP. The child nodes in a tree are distributed according to a Dirichlet process, and maintain Dirichlet process marginals under the described deletion scheme. The posterior probabilities associated with internal nodes are distributed according to a beta distribution; the beta distribution is a special case of the Dirichlet process (where the base measure is atomic with support in two locations), therefore the marginal distribution under deletion remains a beta distribution. □