[Reviews · NeurIPS 2014]

Submitted by Assigned_Reviewer_10

The paper presents a time-varying tree model based on time varying tree structured stick-breaking process. The paper is clear and the model can be useful in a variety of applications where online clustering is required.

Though authors claim the efficiency of their method, computational performance is not reported for any of the application examples. Also, the synthetic data section is weak. It would be nice to compare the proposed algorithm with other approaches on synthetic data. Concerning the real data section, it would be nice to see the qualitative performance of other methods (figure 3 and 4).

Minor comment: figure 3 and 4 are too small.
Summary: Nice paper presenting a model for online clustering. The application section needs to be improved.

Submitted by Assigned_Reviewer_16

This paper introduces the dynamic TSSBP model, a model which ties the stick draws of the TSSBP across time iterations together to give a dynamic hierarchical clustering model. The paper claims that the dTSSBP has TSSBP marginals, however this claim is rather vaguely supported in the supplementary material. Approximate inference is performed in an "online" MCMC update, where only one scan through the data is made. The model is applied to topic modelling on 3 large datasets, with improvements made upon static and non-hierarchical clustering models. The experiments also provide interesting qualitative results.

The paper is well organized and clearly written, however the choice of the form of (2) could be better motivated. The approximate nature of the inference algorithm can be forgiven due to the large dataset applications and the computational difficulty of both hierarchical clustering and time-varying models. The experiments are interesting and clearly demonstrate the benefits of combining hierarchical clustering with time-varying models.

My major qualm is that the proof of stationarity is rather hand-wavy. It seems to me that the claim here is that by marginalizing out z_n^{(s)} for s < t, (2) should reduce to the forms for \nu and \psi in equation (1). This is rather counter-intuitive as in (2) the variables to be integrated out are nonnegative count variables. I strongly recommend that the authors include an explicit proof of their claim in the supplementary material.
Summary: This paper introduces the dynamic TSSBP, a dynamic hierarchical clustering model. This model is applied to several large datasets and improves over dynamic flat models and static hierarchical models in predictive accuracy, and yields some nice quantitative results.

Submitted by Assigned_Reviewer_23

The authors propose a dynamic hierarchical clustering model which allows hierarchies clusters (topics) and corresponding parameters (popularities, word frequencies) to vary in time. Indeed, it is a stochastic process with tree structured marginals so a different hierarchical clustering is specified per time index. Since in general it is a computationally expensive task to do full Bayesian inference they propose an approximate inference scheme where the parameters of each node are MAP estimates and the node re-assignment of observations is done by a Gibbs step. It is based on the tree structured stick breaking process so it allows the observations to be assigned to internal nodes and leaves of the tree rather than just leaves or complete paths. My main concern is that it is not fully exploiting the non-parametric nature of the model since the authors fixed the depth of the trees in the experiments. It would be nice that the depth varied over time or that a more detailed sensitivity analysis was presented.

Some specific comments:

Line 111: In this section I suggest to write the full generative model for each example (this could be added to the supplementary material if it is too long) just to be able to see how one could do full Bayesian inference in this model. Furthermore, maybe a MAP-Bayes like scheme (hard clustering assingment) could be obtained as a limit? I guess this would be a different approach so no need to include this.

Line 244: Here it would be nice to have the corresponding conditional distribution for the parameters \theta for both examples in the previous section. It would be good to have this expression so one can see why you can't obtain MAP estimates directly and hence, the need for the coordinate ascent on this expression.
Line 358: why is it sensitive to initialization? Or is this just done to ensure that the Gibbs sampling has converged?

Line 371: while it is true that shallow hierarchies are easier to interpret it is still not quite satisfying to choose the depth of the tree to be 3 or 4 for every tree. One of the benefits of the model is to get different trees per time index so it would be nice that their corresponding depths varied as well. Furthermore, it would be nice to have a principled way of choosing the depth or at least a criterion which exploits the nonparametric nature of the model (in terms of tree distance, say). For example, as in Lakshminarayanan et al (2013), Mondrian Forests:Efficient Online Random Forests, ArXiv 1406.2673 where they suggest a budget parameter to handle this or in linkage algorithms where you choose your tree depth depending on certain criterion based on a fixed distance.

Line 399: Is it possible to obtain a MAP tree? Since one can obtain a different tree per time index it would be interesting to know if there is a way to combine/summarize such trees just to see if there is a recurring pattern. This could justify or not the assumption that hierarchies vary over time.

Supplementary material:

Line 53: The proof of theorem 1 could be improved. It is not clear to me why is it the case that after deterministic deletion the resulting process is still TSSBP. The proof could be more mathematically precise.

Quality: this is a really well written and structured paper.

Clarity: the importance of the problem is clearly explained and the paper has nice figures to illustrate the concepts and results.

Originality and Significance : the paper introduces novel ideas.

Summary: The authors propose a dynamic hierarchical clustering model which allows topics and corresponding popularities and word frequencies to vary over time and an approximate inference scheme that can be parallelized. They show that their method is better over existing models in terms of predictive performance and recovers reasonable hierarchies per time index. Well written and structured paper.
Author Feedback
Author rebuttal: We thank the reviewers for their encouraging responses, as well as their detailed comments and suggestions. To our knowledge our paper is the first to address the problem of inferring evolving hierarchies in a nonparametric context, maintaining consistent marginals across time (making it appropriate when we do not have a clear “start” and “end” point), and exhibiting self-similar hierarchical clustering (important for incorporating new data in a principled manner). As with other nonparametric tree-based models in the literature, inference is challenging: We believe our approach strikes a good balance between computational feasibility and model integrity, allowing meaningful results on real data.

We address the main questions of the reviewers below.

Reviewer 10:

Computational performance: In line 365, we state that the longest any dTSSBP trial (defined as a single restart/run over all epochs) took was <20 min on a single processor. We also state that some speedup is observed when using 2-4 processors.

Other methods for synthetic data qualitative visualization (Figures 2,3): We did not include comparison figures for the synthetic data for reasons of space; however we are happy to include them in the supplement.

Other methods for real data qualitative visualization (Figures 4,5): Because aligning topics from different models for comparison is difficult, and because our model is the only one that can generate time-evolving hierarchies of topics, it is difficult to compare our Figures 3, 4 against similar visualizations for other models. Hence we decided to leave those out. We would like to point out that our heldout likelihood experiments indicate that dTSSBP’s discovered structure gives a meaningful improvement over other models.

Reviewer 16:

Reviewers 16 and 23 asked for more detail on the proof of TSSBP marginals. We will expand the appropriate section in the appendix. Briefly: the proof is based on a related proof of DP marginals by [1], that relies on the fact that, if \theta ~ DP(alpha, H) and x ~ \theta, the mixture of the \int_xP(\theta|x) P(x|\theta)dx is DP(\theta; alpha, H). The TSSBP is composed of multiple separate DPs (note that the beta distribution is a special case of DP), we can show that each one maintains the appropriate marginals, and therefore the combined marginal is that of the TSSBP by construction.

Reviewer 23:

Line 111: We will incorporate the full generative models into the supplement. We agree that the model should be amenable to a MAP-Bayes inference algorithm; we will explore this in later work.

Line 244: We do have the conditional distributions for \theta, and we will incorporate them into the supplement.

Line 358: The space of possible trees grows exponentially with the number of data points, while the posterior distribution over trees is highly multi-modal, and the topology of the space makes large moves difficult. While, asymptotically, a Gibbs sampler will converge to the correct distribution, the convergence time can be long, meaning in practice our Gibbs sampler is sensitive to initialization. Improving mixing in nonparametric tree models is an ongoing research challenge -- see for example the biostatistics literature on inference coalescent models. While Metropolis Hastings/RJMCMC proposals (e.g. split-merge samplers) may be helpful in improving mixing (and have proved useful in other nonparametric models, see e.g. Jain and Neal, 2003), they are significantly more complicated to implement than Gibbs samplers, and is something we regard as future work. In order to solve the initialization issue while using Gibbs samplers, we resort to random restarts, which are a common strategy in many Bayesian inference applications.

Line 371: For any nonparametric tree, the number of tree nodes (and hence computational cost) grows exponentially with tree depth. Our model and inference algorithm supports trees of unlimited depth, and we limited the tree depth primarily to ensure fast inference (<20 minutes for single run on any of our datasets, as stated in our experiments). We realize that unlimited depth trees are highly desirable, and we are investigating ways to allow unlimited depth without excessive computational complexity. For example, one might limit the branching factor (through the \psi variables), or automatically tune model hyperparameters to favor compact trees (while still supporting unlimited depth and width, if the data necessitates).

Line 399: It was not clear to us if the reviewer is asking about producing a single MAP tree for all epochs, or is interested in finding a MAP tree from multiple samples in the same epoch. In either case, we assume the central issue is sample alignment: given >2 tree samples from the posterior, how do we combine them into one “MAP tree”? While we do not address this issue, we note that Ho et al. [2] describe a relevant procedure in their supplemental, called “consensus samples”: briefly, if two documents are placed in the same hierarchy node in >50% of the samples, then they also put both docs into the same node in the consensus sample. We believe this procedure can be adapted to our model, though we do not know what statistical properties of the tree posterior will be preserved.

Supplementary Line 53: Please refer to our response to Reviewer 16.

[1] F. Caron, M. Davy, A. Doucet. Generalized Polya Urn for Time-varying Dirichlet Process Mixtures. UAI, 2007.

[2] Q. Ho, A. Parikh, L. Song and E. P. Xing. Multiscale Community Blockmodel for Network Exploration. AISTATS, 2011.